# The Aggregation Conditions Define Whether EGCG is an Inhibitor or Enhancer of *α*-Synuclein Amyloid Fibril Formation

**DOI:** 10.3390/ijms21061995

**Published:** 2020-03-14

**Authors:** Rebecca Sternke-Hoffmann, Alessia Peduzzo, Najoua Bolakhrif, Rainer Haas, Alexander K. Buell

**Affiliations:** 1Institute of Physical Biology, Heinrich-Heine-University, 40225 Düsseldorf, Germany; rebecca.sternke-hoffmann@hhu.de (R.S.-H.); alessia.peduzzo@uni-duesseldorf.de (A.P.); Najoua.Bolakhrif@uni-duesseldorf.de (N.B.); 2Department of Hematology, Oncology and Clinical Immunology, Heinrich-Heine-University, 40225 Düsseldorf, Germany; Haas@med.uni-duesseldorf.de; 3Department of Biotechnology and Biomedicine, Technical University of Denmark, 2800 Lyngby, Denmark

**Keywords:** amyloid, EGCG, inhibition, kinetics, seeding, nucleation, Parkinson disease

## Abstract

The amyloid fibril formation by α-synuclein is a hallmark of various neurodegenerative disorders, most notably Parkinson’s disease. Epigallocatechin gallate (EGCG) has been reported to be an efficient inhibitor of amyloid formation by numerous proteins, among them α-synuclein. Here, we show that this applies only to a small region of the relevant parameter space, in particular to solution conditions where EGCG readily oxidizes, and we find that the oxidation product is a much more potent inhibitor compared to the unmodified EGCG. In addition to its inhibitory effects, EGCG and its oxidation products can under some conditions even accelerate α-synuclein amyloid fibril formation through facilitating its heterogeneous primary nucleation. Furthermore, we show through quantitative seeding experiments that, contrary to previous reports, EGCG is not able to re-model α-synuclein amyloid fibrils into seeding-incompetent structures. Taken together, our results paint a complex picture of EGCG as a compound that can under some conditions inhibit the amyloid fibril formation of α-synuclein, but the inhibitory action is not robust against various physiologically relevant changes in experimental conditions. Our results are important for the development of strategies to identify and characterize promising amyloid inhibitors.

## 1. Introduction

The misfolding and uncontrolled aggregation of proteins is linked to the onset and progression of a range of neurological disorders, such as Parkinson’s disease (PD) and Alzheimer’s disease [1,2,3]. PD is a progressive disorder of the nervous system that affects movement, but it also has non-motor symptoms. A pathological characteristic of PD is Lewy bodies [4]. The major filamentous component of Lewy bodies is the presynaptic protein α-synuclein [5,6,7].

α-synuclein is a 140-residue neuronal protein, which can aggregate into highly ordered, cross-β-sheet structured amyloid fibrils [8]. The aggregation mechanism of α-synuclein is a complex nucleation-dependent polymerization process, which manifests itself in test tube experiments of aggregation kinetics through a lag phase, followed by a growth and a steady-state phase [9]. At a molecular level, this behavior stems from the combination of different microscopic steps, in particular (heterogeneous) primary nucleation and growth and, depending on the solution conditions, fragmentation and secondary nucleation [3,10].

At neutral pH α-synuclein carries a net negative charge, due to its isoelectric point of 4.0–4.7 [11]. The protein adopts a primarily disordered conformation without a well-defined secondary or tertiary structure (intrinsically disordered conformation) [12,13]. At low pH, α-synuclein displays a more compact conformation due to a collapse of the normally highly acidic and extended C-terminal tail [14]. The C-terminal tail becomes fully protonated, thus uncharged, and it forms a compact conformational ensemble with increased hydrophobicity, resulting in an accelerated α-synuclein fibril formation at low pH [10]; a similar effect can be achieved by truncating the C-terminus [15].

Compared to aggregation at neutral pH, the aggregation process is strongly enhanced at mildly acidic pH values (pH < 6), through an efficient production of new growing fibrils catalyzed by the binding to the surfaces of pre-existing fibrils (secondary nucleation) [10,16]. The investigation of the behavior of α-synuclein at mildly acidic pH values is physiologically relevant, because α-synuclein can experience such solution conditions during its life cycle in endosomes and lysosomes, which maintain an acidic pH value between 4 and 5 [17,18,19]. This pH range has to be considered when designing a possible therapeutic strategy for the treatment of synucleinopathies.

Once formed, amyloid aggregates are often found to be very stable in vitro [20], even though examples of less stable fibrils have been found [21]. In general, it is not straightforward to compare the stability of monomeric proteins with that of amyloid fibrils, because the latter is concentration-dependent [22]. In vivo, it appears to be difficult in many cases to reverse the aggregation process and to prevent it from spreading through secondary processes. Therefore, the development of inhibitors that are able to prevent the initial formation of amyloid aggregates is crucial. In particular, a target of these substances are small oligomeric species, which are suspected to be the most cytotoxic [23]. Targeting these oligomers has therefore been proposed to be an efficient strategy to combat neurodegenerative diseases. Viable therapeutic strategies for prevention and treatment of amyloid-related diseases are (1) the maintenance of the amyloidogenic protein in its soluble state, (2) the redirection of the amyloidogenic proteins into unstructured, nontoxic, and off-pathway aggregates, and (3) remodeling and/or dissociation of the mature amyloid fibrils. One class of inhibitors of α-synuclein amyloid fibril formation is formed by molecules that can interact with amyloidogenic monomers in order to prevent their inter-molecular associations, such as antibodies or affibodies [24,25]. Small molecules have also been proposed as inhibitors of α-synuclein amyloid fibril formation [26,27,28,29]. These include polyols, polyphenols, or other aromatic molecules containing hydrogen-bonding functionalities.

Polyphenols are naturally occurring secondary metabolites of plants characterized by the presence of two or more phenol rings [30]. A well-studied polyphenol is epigallocatechin-3-gallate (EGCG), the main polyphenol found in green tea. Biochemical studies indicate the neuroprotective action of EGCG, which has been suggested to inhibit the aggregation of a number of amyloidogenic peptides and proteins effectively, including α-synuclein [31,32,33,34,35], amyloid-β (related to AD) [31,36], islet amyloid polypeptide (related to type-II diabetes) [37,38], huntingtin exon 1 (related to Huntington’s disease) [39], tau (related to AD and tauopathies) [40], superoxide dismutase (related to amyotrophic lateral sclerosis) [41], prion proteins (related to prion diseases) [42,43], and others. EGCG has the ability to prevent the formation of toxic prefibrillar oligomers, as well as to inhibit amyloid fibril formation and has been proposed to remodel existing amyloid fibrils. Most of the studies are performed at physiological pH, where EGCG is unstable and oxidizes rapidly into various products [44], which we collectively denote as EGCG_*ox*_ in this work. Decreasing the pH to slightly acidic pH values (≤pH 6) results in a considerable increase in EGCG stability, which we have recently shown to lead to a strongly impaired ability of EGCG to inhibit α-synuclein aggregation [45].

Considering that both α-synuclein and EGCG are likely to encounter such pH environments in vivo, the aim of the present study is to characterize in more detail the dependence of EGCG inhibition of α-synuclein aggregation on the solution conditions. The in vitro amyloid fibril formation of α-synuclein is an intrinsically slow process, due to the fact that the primary nucleation process is heterogeneous and requires a combination of appropriate surfaces (e.g., air-water interface [46], polymer-water interface [47], or lipid-water interface [48]) and constant agitation of the sample. The nucleation step can be bypassed by the addition of exogenous fibrils (seeds). Seeding, in particular at high seed concentrations (where all molecular processes other than fibril growth can be neglected [49]), can improve the reproducibility of the aggregation kinetics of α-synuclein. In this study, we pay special attention to how the effect of EGCG on the fibril formation is altered, when the aggregation of α-synuclein is studied under different conditions, such as different surfaces and the presence and absence of glass beads. By systematically examining a wide range of pH values, we probe the influence of EGCG oxidation and the interplay between the solution conditions and the action of EGCG.

Overall, we find that EGCG is an efficient inhibitor of α-synuclein aggregation only in a small range of the parameter space. Under some conditions, especially those that lead to enhanced stability of EGCG, it can either have no effect on the amyloid fibril formation of α-synuclein or even promote the latter through facilitating heterogeneous primary nucleation.

## 2. Materials and Methods

### 2.1. Materials and Solutions

α-synuclein in the pT7-7 vector was expressed in *Escherichia coli* BL21 (DE3) and purified as previously described [50]. As a last step, α-synuclein was purified by size-exclusion chromatography on an ÄKTA pure chromatography system (GE Healthcare, Chicago, IL, USA) using a Superdex 200 Increase 10/300 GL (GE Healthcare) and 20 mM citric acid, pH 7, as an elution buffer. α-synuclein concentration was determined by measuring UV-absorption at 275 nm (extinction coefficient of 5600 M^−1^ cm^−1^). For the α-synuclein inhibition experiments, 5 mM solutions of EGCG (Tocris #4524, Bristol, UK) were prepared by dissolving EGCG in dH_2_O. The EGCG solutions were frozen and stored at −20 °C, after observing no difference between freshly dissolved and thawed EGCG. EGCG_*ox*_ was prepared by dissolving 10 mM of EGCG in 20 mM citric acid, pH 7, and incubating for at least 6 h at 60 °C in a Thermomixer Compact (Eppendorf, Hamburg, Germany) at 1000 rpm. Subsequently, it was diluted to a final concentration of 5 mM, frozen, and stored at −20 °C.

### 2.2. Measurements of Aggregation Kinetics

In order to study the effect of EGCG on the amyloid fibril formation by α-synuclein, solutions of 25 μM of α-synuclein were prepared with EGCG or EGCG_*ox*_ solutions in a 1:1 and 1:5 (protein:compound) ratio, 20 μM ThT, and 150 mM citric acid at the desired pH value (pH 3, pH 4, pH 5, pH 6, or pH 7). Three replicates of each solution were then pipetted into a high-binding surface plate (Corning #3601, Corning, NY, USA) or a non-binding surface plate (Corning #3881). The aggregation kinetics were monitored in the presence and absence of small glass beads (SiLibeads Type M, 3.0 mm). The plates were sealed using SealPlate film (Sigma-Aldrich #Z369667, St. Louis, MO, USA). The kinetics of amyloid fibril formation were monitored at 37 °C under continuous shaking (300 rpm) or quiescent conditions by measuring ThT fluorescence intensity through the bottom of the plate using a FLUOstar (BMG LABTECH, Ortenberg, Germany) microplate reader (readings were taken every 5 min).

In order to investigate the interactions of EGCG and the surface of the protein-repellant (non-binding) surface plate, 130 μL of the solutions of EGCG or EGCG_*ox*_ (25 μM and 125 μM) were pipetted into a well and incubated at room temperature for 2 h. After incubation, the solutions were removed, the concentrations of EGCG and EGCG_*ox*_ were measured by UV-absorption, and a solution of 25 μM α-synuclein, 20 μM ThT, and 150 mM citric acid pH 4 was added to the pre-treated wells. Three replicates per condition were measured. The EGCG_*ox*_ concentration was compared to a solution that was incubated in an Eppendorf tube. The highest ThT fluorescence emission value within each time course was taken to be I_*max*_. The half-times (t_50_) of the aggregation reaction were obtained as described by Meisl et al. [51].

### 2.3. Seeded Aggregation Experiments

In order to probe the effects of EGCG and EGCG_*ox*_ specifically on the elongation process and on preformed fibrils, the aggregation kinetics of α-synuclein were monitored in the presence of 5% and 0.5% seeds in relation to the used monomer concentration. For the preparation of the seeds, solutions of 50 μM α-synuclein were incubated at 37 °C in a high-binding surface plate, in the presence of glass beads and under continuous shaking at the desired pH-values (pH 3, pH 4, pH 5, pH 6, or pH 7). The end product after 24 h of incubation, when the reaction was found to be completed under all conditions, was used to seed fresh solutions of 50 μM α-synuclein, which were incubated at 37 °C for 24 h in an Eppendorf Thermomixer with continuous shaking (1200 rpm). The product of this seeded aggregation experiment was used to seed the kinetic experiments in the presence and absence of EGCG and EGCG_*ox*_. For the calculation of the seed concentration, it was assumed that the monomer had quantitatively converted to amyloid fibrils. Before performing the experiment, the seed-solutions were homogenized using an ultra-sonication bath Sonorex RK 100 H (Bandelin, Berlin, Germany) for 180 s. The seeded aggregation experiments were performed in non-binding surface plates with 25 μM α-synuclein monomer, and the preformed fibrils were added as seeds to a final concentration of 0.5 or 5% of the monomer solution at the desired pH value. For the experiment with seeds that were to be pre-incubated with EGCG, we added 50 μM EGCG or EGCG_*ox*_ to a solution of 50 μM α-synuclein fibrils, at pH 5, pH 6, and pH 7. After 2 h, the fibrils were added to 25 μM monomer at the corresponding pH value to a final concentration of 5%, and the aggregation kinetics were recorded every 150 s. In order to investigate if EGCG could remodel pre-formed fibrils, we incubated 10 μM fibril solutions, which were sonicated beforehand, with 10 μM EGCG or EGCG_*ox*_ in a non-binding surface plate at 37 °C under shaking conditions at pH 4, pH 6 and pH 7 for over 100 h. The fluorescence intensity was recorded using a FLUOstar (BMG LABTECH) microplate reader (readouts were taken every 5 min). After the incubation, 50 μM fresh monomer was added to the solutions, and the measurement was continued, with readings taken every 150 s, allowing a potential change in the seeding efficiency to be detected.

### 2.4. Atomic Force Microscopy

AFM images were acquired directly after the aggregation kinetic measurements. Ten microliters of each sample were deposited onto freshly cleaved mica. After drying, the samples were washed 5 times with 100 μL of dH_2_O and dried under a gentle flow of nitrogen. AFM images were obtained using a NanoScope V (Bruker, Billerica, MA, USA) atomic force microscope equipped with a silicon cantilever ScanAsyst-Air with a tip radius of 2–12 nm.

### 2.5. Microfluidic Diffusional Sizing and Concentration Measurements

Fluidity One (F1, Fluidic Analytics, Cambridge, UK) is a microfluidic diffusional sizing (MDS, [52]) device that measures the rate of diffusion of protein species under steady state laminar flow and determines the average particle size from the overall diffusion coefficient. The protein concentration is determined by fluorescence intensity, as the protein is mixed with ortho-phthalaldehyde (OPA) after the diffusion, a compound that reacts with primary amines, producing a fluorescent compound [53]. To measure the concentration of the soluble α-synuclein, the samples were centrifuged for 60 min at 16,100× *g* at 25 °C using Centrifuge 5415 R (Eppendorf) directly after the kinetic measurements. The top half of the supernatant was removed, and 6 μL of this solution were pipetted onto a disposable microfluidic chip and measured with the Fluidity One. For the measurement of the amount of protein in the pellet, the pellet was re-suspended in the remaining liquid, and 6 μL of this solution were pipetted onto a microfluidic chip and analyzed with the F1 instrument.

## 3. Results

### 3.1. Non-Seeded Experiments in High-Binding Plates

We performed α-synuclein amyloid fibril formation experiments in high-binding surface plates in the presence and absence of glass beads

(Figure 1). α-synuclein monomer solutions are kinetically highly stable, because the homogeneous primary nucleation rate is slow, in particular around neutral pH. Under most conditions, the primary nucleation of α-synuclein is triggered by surfaces that have an affinity for the protein. When α-synuclein binds to a suitable surface or interface, in particular lipid bilayers, the N-terminal domain can adopt a helical structure [54], which seems to facilitate the formation of fibril nuclei and oligomers [48]. The air-water interface [46], the surface of the plate [47], and under mildly acidic pH conditions, the surface of already formed fibrils [10,16,50] can also function as a nucleation assistant. The growing fibrils can then be fragmented by shaking, in particular in the presence of glass beads [10], which leads to accelerated aggregation kinetics.

α-synuclein monomers were combined with the compound EGCG or EGCG_*ox*_ in a protein:compound ratio of 1:1 (25 μM:25 μM) or 1:5 (25 μM:125 μM), and the aggregation was monitored through the change in thioflavin-T (ThT) fluorescence intensity. An increase in the intensity indicated the formation of ThT-positive aggregates, specifically amyloid fibrils. The results of these initial experiments in polystyrene plates are shown in Figure 1, and the extracted maximal fluorescence intensities and aggregation half times are shown in Figure 2. The half time (t_50_) was the time point when the ThT intensity had reached half the value between the initial baseline and the final plateau value and could be used as a macroscopic parameter to describe the aggregation kinetics. α-synuclein displayed the fastest aggregation kinetics at pH 4 with a t_50_ of 1.1 h when glass beads were added. Without glass beads, the fibril formation was slowed down by a factor of two. The de novo aggregation at pH 3 and pH 5 was similar, and the kinetics under these two conditions were slower by a factor of 2–3 (with added glass beads) and 8 (without additional glass beads), compared to pH 4. The aggregation at pH 5 was slightly faster than at pH 3, and the influence of the glass beads on the kinetics was bigger. The glass beads had the strongest aggregation enhancing influence at pH 7. α-synuclein could not form amyloid fibrils very efficiently at pH 7, possibly due to the high negative charge under these conditions, which was not strongly screened at the moderate ionic strength values of our experiments. Since secondary nucleation was suppressed at neutral pH [10], the dominant molecular events were primary nucleation and elongation of the newly formed fibrils. In the presence of glass beads, the enhanced fragmentation led to a t_50_ of 9 h, whereby the absence of glass beads led to an almost complete absence of secondary processes and a t_50_ of 96 h.

When the maximal ThT fluorescence intensities were compared to I_*max*_ of the control, i.e., the absence of either EGCG or EGCG_*ox*_, almost all conditions showed a decrease, in particular at neutral and slightly acidic pH (pH 7–pH 6) (Figure 2). Only the aggregation at pH 3 showed an increased fluorescence intensity in the presence of EGCG. In the presence of EGCG_*ox*_, the intensities were decreased under all conditions indicating either an inhibitory effect on the aggregation or an interference with the ThT-signal. When the experiments were performed without glass beads, I_*max*_ showed a similar outcome.

If rather than I_*max*_, the half-time of the aggregation reaction, t_50_, was used as a read out, the observed inhibitory effects of EGCG at neutral and slightly acidic pH (pH 7 and pH 6) were different. At pH 7 and in the presence of EGCG, there was no increase in ThT-signal observable in the presence of glass beads, whereas there was a slight increase in fluorescence intensity over time in the absence of glass beads. In one of the three repeats in the presence of EGCG_*ox*_, a minor fluorescence intensity increase was also observed in the case of the equimolar ratio. The t_50_ of the aggregation in the presence of EGCG was clearly increased at pH 7. The picture at pH 6, however, was different. Where in the presence of EGCG (1:1), the intensity was significantly (*p* ≤ 0.01) decreased, the t_50_ was not distinguishable from the control. In the presence of a high concentration of EGCG, the kinetics were faster both with and without additional glass beads. However, the ThT-intensity curves without glass beads did not show the typical sigmoidal shape. The decrease of the signal at longer times could be explained by higher order assembly or surface adsorption of the amyloid fibrils [55], or else a time-dependent interference of EGCG with ThT fluorescence. In the presence of EGCG_*ox*_, the kinetics were prolonged and even completely inhibited, both with and without glass beads. However, quantification of the soluble protein in the supernatant at the end of the experiment with additional glass beads showed a loss of 62–67% in the presence of EGCG_*ox*_ (1:1) and (1:5), compared to the starting monomer concentration. The formed aggregates were accordingly either not ThT-positive, or else, the EGCG interfered strongly with their ThT fluorescence.

At more acidic pH values, only the oxidized form of the EGCG showed any clear effect on the aggregation kinetics. At pH 5 and pH 4, the EGCG_*ox*_ (1:5) showed a significant increase of t_50_, but in the absence of glass beads, the EGCG showed an accelerating effect at pH 5 and pH 3. At pH 4, the aggregation kinetics were slowed down slightly, but not as much as in the presence of glass beads, and it was not statistically significant. The variability in the kinetics of the three replicates per condition, especially at pH 3 and pH 5, was high, rendering a thorough statistical analysis difficult with such a small number of replicates.

### 3.2. Non-Seeded Experiments in Non-Binding Plates

The amyloid formation of α-synuclein was favored in high-binding surface plates, due to the ability of the polystyrene-water interface to provide nucleation sites. In an attempt to try and disentangle the effects of EGCG and EGCG_*ox*_ on the polystyrene surface-induced nucleation from that on other relevant processes, we also performed aggregation experiments in non-binding surface plates, i.e., plates where the surface was coated with a protein-repellent PEG layer (Figure 3). These non-binding surface plates are often used in kinetic experiments of amyloid fibril formation in order to minimize the contribution of heterogeneous nucleation processes and therefore simplify the kinetic analysis [10,48,56]. As expected, therefore, the kinetics of de novo aggregation in the non-binding surface plates were slowed down with respect to the high-binding plates, but the high fragmentation rate in the presence of glass beads allowed fibril formation to be observed within an experimentally accessible time scale. The enhanced fragmentation amplified the fibrils that formed at the air-water interface [46]. This difference in lag times between high and low binding plates was particularly pronounced at the pH values that led to very fast aggregation kinetics in the high-binding surface plates: at pH 4, the t_50_ was increased by a factor of 3.4, whereas it was only increased by a factor of 1.3 at pH 7.

While the maximal ThT-intensities in the presence of glass beads and EGCG or EGCG_*ox*_ in non-binding plates were comparable to the aggregation experiments conducted in a high-binding surface plate, the t_50_ values indicated an aggregation-enhancing effect of both oxidized and fresh EGCG (Figure 4).

The enhanced fragmentation of the fibrils by the use of glass beads lead in general to more reproducible data, both in binding and non-binding plates. However, the aggregation kinetics under some conditions (e.g., pH 6) were still rather variable, as both I_*max*_ and t_50_ differed between the three replicates of the control sample. In order to obtain an independent (of ThT fluorescence intensity) measurement of aggregate mass in this case, we centrifuged the samples at the end point of the experiments and measured the size and concentration of the soluble α-synuclein in the supernatant (Figure 5A, left panel) by microfluidic diffusional sizing (MDS) [52]. We found that while the ThT-signal displayed a clear difference, the amount of soluble protein in the supernatant was similar. In all three samples, α-synuclein converted near-quantitatively into aggregates, and the average size of the supernatant fraction was that of monomeric protein. Overall, the picture that emerged from these ThT experiments in non-binding plates with added glass beads was that EGCG and EGCG_*ox*_ only had an inhibitory effect at pH 7 and in addition at pH 6 in the presence of a five-fold excess of EGCG_*ox*_. This conclusion was confirmed by MDS experiments under all conditions (Figure 5A, central panel), which showed that despite the variable final ThT intensity, the protein was quantitatively converted into fibrils, illustrating that EGCG and EGCG_*ox*_ could have a strong influence on ThT intensity [45].

In the absence of glass beads, the aggregation curves were generally more variable also in the non-binding surface plate. The absolute fluorescence intensities at the plateau were increased by a factor of up to almost 10 in the presence of EGCG (for example, at pH 6 with EGCG and EGCG_*ox*_ (1:1) and pH 5 with EGCG_*ox*_ (1:1)), and only in the presence of the five-fold excess of EGCG_*ox*_ could we detect similar or slightly decreased maximal intensities compared to the control. Importantly, the aggregation kinetics in the absence of glass beads were accelerated with respect to the control in the presence of both oxidized and fresh EGCG, in particular by the former. This accelerating effect was particularly pronounced at acidic pH (pH 3 and 4), but also observed at pH 5 and 6. Again, only at pH 7, an inhibitory effect was observed. The control aggregation (no EGCG) at pH 7 in the absence of glass beads displayed a peculiar characteristic: the ThT fluorescence increase started already after ca. 10 h, but the rate of increase in fluorescence intensity was slow. The analysis of the supernatant after more than 100 h demonstrated that over 80% of the protein was still soluble and had an average radius of 3.12 nm (Figure 5B), corresponding to the expected size of the monomer [57]. The lag time of the sample with an equimolar concentration of EGCG was slightly longer, but the reaction reached a much higher final fluorescence level. All the other samples at pH 7 without glass beads did not show any fluorescence intensity increase. We also probed for the presence of fibrils and the degree of aggregation under those conditions with the help of AFM imaging and MDS. In these MDS experiments, we measured both the pellet and the supernatant and calculated the total concentration for which the MDS data could account. Based on the observation that the combined concentrations in both pellet and supernatant did not add up to the initially used concentration, we concluded that the samples with 1:5 EGCG and those at 1:1 and 1:5 EGCG_*ox*_ also aggregated, but the aggregates were not ThT-positive (either due to their non-fibrillar nature or due to quenching by EGCG) and were probably too big to be quantifiable by MDS. This was compatible with the observation that in all samples, except for the one with 1:1 EGCG, the α-synuclein was still primarily monomeric with radii between 2.4 and 3 nm.

At pH 4, in non-binding plates and in the absence of glass beads, all EGCG and EGCG_*ox*_ containing samples showed faster and more reproducible aggregation than the control samples. In order to probe whether the type of aggregates formed under all these conditions was the same, we performed time-resolved AFM imaging experiments. Aliquots were taken at different time points directly out of the plate during the measurement (Figure 3B) and imaged by AFM (Figure 6). The sample of α-synuclein with a five-fold excess of EGCG_*ox*_ (1:5) (magenta frame) displayed many short fibrils and some amorphous structures after 17 h, when the ThT-signal had already reached the plateau-phase for several hours. For the other samples, we only found fibrils under all conditions in the last time point. At 17 h and 42 h, for example, we could not find any fibrillar aggregates in the control and 1:1 EGCG_*ox*_ samples, even if the increase in ThT-signal suggested fibril formation, highlighting the fact that imaging-based analysis alone could be unreliable in some cases. Overall, we found no clear difference in appearance of the fibrils made under any of the different EGCG regimes at pH 4, confirming that neither EGCG nor EGCG_*ox*_ displayed an inhibiting effect under these conditions and that the observed accelerated emergence of ThT fluorescence could indeed be ascribed to an enhancing effect of the EGCG and EGCG_*ox*_.

As described above, α-synuclein normally requires an appropriate surface or interface to induce the nucleation of its amyloid fibrils. Therefore, the question arises as to how EGCG and EGCG_*ox*_ are able to accelerate the formation of amyloid fibrils in non-binding surface plates. It could be that the compounds directly interact with the monomeric α-synuclein and facilitate nucleation, or else, the compounds interact with the non-binding plate surface and render it conducive to induce α-synuclein amyloid fibril nucleation. We hypothesized that if EGCG and EGCG_*ox*_ had a high affinity for the non-binding plate surface, pre-treatment of the plate with the compounds should have a comparable effect as if the compounds were present during the entire aggregation experiment. Therefore, we incubated wells for two hours at room temperature with solutions of 25 μM (corresponding to a 1:1 ratio) or 125 μM EGCG or EGCG_*ox*_ (corresponding to a 1:5 ratio). After the incubation, the solution was removed, and the concentration of EGCG in the removed solution was determined. We found that the concentration of the 25 μM EGCG_*ox*_ solution was decreased by 20% and of the 125 μM EGCG_*ox*_ solution by 28% compared to the solutions that were incubated in an Eppendorf tube for the same time duration. The loss of ca. 6 μM from the 1:1 EGCG solution was compatible with the formation of a monolayer of EGCG on the surface of the well, assuming that one molecule could occupy a surface area of approximately 1 nm × 1 nm. It is interesting that at the higher concentration, the EGCG solution is depleted by an approximately proportional amount, which could suggest the formation of supramolecular EGCG structures either in solution or at the surface. Alternatively, it could also be explained by a weak binding affinity that led to saturation of all surface binding sites only at concentrations much higher than the ones used here. Aggregation in the wells treated in this way was indeed found to be accelerated in many cases with respect to the control (Figure 7). In particular, the pre-treatment with EGCG_*ox*_ was found to be an efficient way to enable the plate to induce α-synuclein aggregation. Most notably, the surface pre-treated with 125 μM EGCG_*ox*_ was even more efficient (t_50_ of 4.2 h ± 0.33 h) than if the same concentration of 125 μM EGCG_*ox*_ was present during the aggregation reaction (t_50_ of 4.6 h ± 0.74 h). In the case of 25 μM EGCG_*ox*_, pre-treatment was found to be less efficient (t_50_ of more than 120 h) than the presence of the compound (t_50_ of 10.5 h ± 1.14), suggesting that 25 μM EGCG_*ox*_ may not be enough to saturate the surface during pre-treatment. This conclusion was in agreement with the one drawn from the EGCG depletion experiments described above. These experiments were performed under quiescent conditions, explaining why some reactions displayed a slower kinetics compared to the equivalent solution conditions in Figure 3B (control and EGCG 1:1 and 1:5). After the aggregation experiment, we centrifuged the samples and quantified the average size and concentration of the soluble protein by MDS. The samples that did not display an increase in the ThT-fluorescence had indeed remained mostly soluble, whereas the samples with EGCG_*ox*_ aggregated nearly completely. The intensity I_*max*_ of the samples in pre-treated wells was higher than the corresponding ones with the same concentrations of EGCG in solution, even though the samples in the wells pre-treated with 25 μM EGCG and EGCG_*ox*_ contained still a small amount of soluble protein. We could confirm the presence of fibrils of α-synuclein in the pre-treated wells by AFM. The sample with EGCG_*ox*_ (1:5) present, which corresponded to the time resolved AFM sample in Figure 6, showed mostly amorphous material this time, even though we saw fibril formation (Figure 3 and Figure 6) with almost identical kinetic traces. This variability in imaging, but not in kinetic traces illustrated the fact that AFM imaging is not always representative of the distribution of species in the solution. In addition, the chemical nature of any structure was difficult to ascertain by AFM. The amorphous material in Figure 6 could be protein, but also could correspond to the EGCG content of the solution.

### 3.3. Seeded Experiments in the Presence of EGCG and EGCG_*ox*_

In all the experiments described above, the nucleation and growth of α-synuclein amyloid fibrils proceeded simultaneously, in some cases (pH < 6) in combination with secondary nucleation. A common strategy in the mechanistic analysis of protein aggregation is to perform seeded experiments, which can strongly accelerate the overall aggregation time course and which, at high enough seed concentration, allow studying the process of fibril elongation in isolation [10,49]. Experiments at weaker seeding can be useful if the contribution of secondary processes, such as fibril fragmentation or surface-catalyzed secondary nucleation, are to be studied [10]. In the case of α-synuclein, seeded experiments are often performed under quiescent conditions in a non-binding surface plate, in order to minimize the de novo formation of fibrils. In order to investigate the effects of EGCG and EGCG_*ox*_ on the elongation process, we added 5% seeds to monomeric α-synuclein with and without a five-fold excess of EGCG and EGCG_*ox*_ with respect to the concentration of monomeric α-synuclein at different pH values (Figure 8). We recently proposed a definition of a measure for the seeding efficiency of a given batch of seed fibrils, based on the analysis of strongly seeded kinetic data [49]. A seeding efficiency of one seeding unit (s.u.) corresponded to an effective exponential constant of 1 h^−1^ under conditions where the normalized kinetic traces of the seeded experiments could be well-fitted by the function 1 − e^−*kt*^, with the time t in hours. We quantified the following seeded experiments within this framework, allowing for a convenient comparison of the effects of EGCG and EGCG_*ox*_. It has to be kept in mind, however, that in this framework, the effects of the inhibitor were entirely attributed to their action on the seed fibrils, and interactions with the soluble protein were not included. Therefore, this framework was most appropriate for the experiments with pre-incubated seeds.

Fresh EGCG did not show an inhibitory influence on the seeding efficiency, but the fluorescence intensities were decreased in a pH-dependent manner compared to the absence of EGCG. At both pH 3 and 6, the seeding efficiency in the presence of EGCG appeared even to be somewhat increased with respect to the control. The strongly seeded aggregation kinetics of α-synuclein at pH 7 in the presence EGCG appeared unusual. After a fast increase, the ThT fluorescence intensity decreased strongly. However, by AFM, we were able to verify the presence of fibrillar structures. The fibrils were present as big aggregates on the mica substrate (Figure 8A), which was somewhat unexpected at neutral pH, given that higher order assembly of α-synuclein fibrils was most pronounced at pH values close to the isoelectric point [10]. The observed decrease in ThT signal could be due to this higher order assembly of the fibrils or alternatively to the fact that the initial seeded aggregation was faster than the EGCG oxidation, but the latter will ultimately be responsible for a decrease in fluorescence intensity through quenching by the EGCG_*ox*_. Due to this unusual shape, we did not quantify the seeding efficiency in the presence of EGCG at pH 7. Furthermore, the seeded aggregation at pH 4, both in the presence and absence of EGCG, showed a biphasic behavior, indicative of a contribution of secondary processes, also preventing the application of the simplified framework for the determination of seeding efficiencies. Such behavior was unexpected at high seed concentrations, but could be explained by higher order assembly, which was particularly pronounced close to the isoelectric point [10] and which decreased the seeding efficiency. However the similarity of the kinetic traces indicated that also at pH 4, EGCG had no inhibitory effect on the seeding efficiency. The situation was quite different for 1:5 EGCG_*ox*_, which dramatically reduced the seeding efficiency and led to sigmoidal aggregation curves at pH 3–5 even at this high seed concentration of 5%. The observed lag times were even longer than in experiments without added seeds in non-binding plates (Figure 9C). However, these experiments could not be compared in a straightforward manner, because the non-seeded experiments were performed and the seeded experiments quiescently. Nevertheless, this result suggested that at the most acidic pH values (3–5), EGCG_*ox*_ inactivated the pre-formed seeds, and the observed ThT intensity increase was due to de novo formation of fibrils. AFM measurement confirmed the presence of fibrils in the samples with EGCG_*ox*_ under all pH conditions. The fitting of the kinetics, which showed the expected shape for strongly seeded experiments (i.e., single exponential function, pH 6–7), revealed a somewhat decreased seeding efficiency with respect to the control (Figure 9A).

When less seeds are added to the experiments (0.5% seeds in monomer equivalents), the aggregation kinetics were considerably slower than at the 10-fold higher seed concentration of the previous experiments (Figure 9B), potentially allowing the impact of EGCG and EGCG_*ox*_ on secondary processes to be studied. An inhibitory effect of both EGCG and EGCG_*ox*_ was clearly visible at pH 7. The aggregation started immediately, without a lag phase, in the control sample, with EGCG_*ox*_ after over 30 h, and in the presence of EGCG, no increase in ThT signal was observed after over 60 h. Compared to the aggregation with 5% seeds, EGCG was therefore found to have a stronger impact on the aggregation kinetics at the decreased seed concentration. EGCG_*ox*_ also delayed the aggregation reaction at pH 5 and pH 6, whereas it had an accelerating effect at pH 4. At the latter pH, the weakly seeded data in the absence of EGCG or EGCG_*ox*_ had the expected sigmoidal shape indicative of secondary nucleation. The aggregation kinetics in the presence of EGCG_*ox*_ resembled that in the absence of seeds (Figure 3 and Figure 7), where the EGCG_*ox*_ induced de novo formation of fibrils by changing the properties of the non-binding plate surfaces. Overall, it appeared that in these seeded experiments in non-binding plates, several competing effects were at work, all of which were pH dependent. Fresh EGCG mostly exerted its inhibitory effect at pH 7, consistent with the non-seeded experiments. EGCG_*ox*_, on the other hand, was able to effectively interfere with seeded aggregation at acidic pH values, while at the same time being able to render the non-binding plate conducive to nucleate α-synuclein amyloid fibrils.

### 3.4. Experiments in the Presence of Seeds Pre-Incubated with EGCG and EGCG_*ox*_

In order to separate the effects of EGCG on the seeds and on the soluble α-synuclein, we also performed experiments where we incubated seed fibrils for 2 h at room temperature with stoichiometric amounts of EGCG and EGCG_*ox*_ and added them to a final concentration of 5% (in monomer equivalents) to 25 μM monomeric α-synuclein. This corresponded at the same time to a strong dilution of the EGCG, such that the ratio of soluble α-synuclein to EGCG/EGCG_*ox*_ was 20:1 during the seeded experiments. If the EGCG either bound with high affinity to the fibrils and/or was able to remodel the fibrils into non-fibrillar structures [31], then a reduction in seeding efficiency could be expected. However, the observed aggregation kinetics were very similar, particularly when they were normalized to the same final level of fluorescence intensity (Figure 10A). A quantitative analysis showed that the seeding efficiency showed no significant difference between the samples, suggesting that during the time scale of this experiment, the fibrils had not undergone a significant structural change. In order to investigate whether fibril remodeling into seeding-incompetent species could occur over longer time scales, we incubated 10 μM pre-formed seeds in a non-binding surface plate at 37 °C for over 100 h at pH 4, 6, and 7 with equimolar concentrations of EGCG or EGCG_*ox*_ (Figure 10B). The ThT-intensity of the control did not change over time, but the ThT intensity of the fibrillar sample with added EGCG decreased by a factor of 2.4 at pH 6 and 1.8 at pH 7, whereby the decrease at pH 6 was slower. The fluorescence intensity in the presence of oxidized EGCG was reduced by a factor of 2.2 at pH 6 and 2.1 at pH 7. The intensities of the samples with both EGCG and EGCG_*ox*_ were very similar; the EGCG_*ox*_ did not lead to a stronger quenching of the fluorescence. Compared to the control, the intensities after over 100 h were lowered by a factor of 3.2 at pH 6 and 4.2 at pH 7 in the presence of EGCG and 4.7 in the presence of EGCG_*ox*_. Despite the fact that the significant decrease in ThT fluorescence intensity suggested a change of the seed fibril structure or concentration, and correspondingly a different seeding efficiency, the observed kinetics after addition of 50 μM monomer were virtually identical between all samples. We quantified the seeding efficiencies and found no statistically significant differences induced by the long incubation with EGCG or EGCG_*ox*_. Therefore, an equimolar concentration of EGCG was not able to induce changes in the seeding efficiency of preformed fibrils even after prolonged incubation.

## 4. Discussion

The effects of the potential anti-amyloid component EGCG and its oxidation products on the process of amyloid fibril formation of the protein α-synuclein was analyzed under distinct environmental conditions. We probed the effect of pH in the range from pH 3 to pH 7, the effect of the presence of glass beads, the influence of the type of plate surface, as well as the presence and absence of seeds, which were either freshly prepared or pre-incubated with EGCG or EGCG_*ox*_. By examining the change of the maximum ThT fluorescence intensity and/or the kinetics of the aggregation (quantified by the t_50_ of the reaction) in the presence of the compound, the effects on the de novo (i.e., unseeded) amyloid formation were assessed, and a summary is presented in Table 1.

The overall picture that emerges from Table 1 is that only at pH 7, both I_*max*_ and t_50_ suggested an inhibitory effect of EGCG under all tested conditions. This was confirmed by microfluidic diffusional sizing experiments, which showed that α-synuclein was maintained in its a soluble, probably monomeric state by both EGCG and EGCG_*ox*_. Already at pH 6, the picture became more complex, and the two parameters I_*max*_ and t_50_ did not yield a consistent picture: while I_*max*_ still largely suggested inhibition, t_50_ showed mostly no effect of EGCG, but still inhibition for EGCG_*ox*_. At neutral pH (i.e., pH 7), corresponding to the solution condition most often investigated in previous studies, EGCG was highly unstable. EGCG oxidized rapidly under these conditions within a similar time scale as the aggregation process itself in the presence of glass beads [45]. Despite the fact that the simultaneous occurrence of α-synuclein aggregation and EGCG oxidation at pH 7 rendered a detailed mechanistic analysis challenging, it was important to include these conditions in the present study, given that they corresponded to the most widely employed and studied solution conditions [31,58]. A decrease of the pH value to pH 6 led to a significant increase in the stability of EGCG, allowing the aggregation process to be de-coupled from the process of EGCG oxidation. Accordingly, a less strong inhibitory effect on α-synuclein aggregation was found at pH 6 and at even lower pH values, where EGCG was highly stable, and no significant effects on the de novo amyloid fibril formation of α-synuclein were observed. Aggregation experiments performed in the presence of both EGCG and EGCG_*ox*_ at pH values where no detectable oxidation of EGCG occurred allowed the conclusion that mostly EGCG_*ox*_ was able to inhibit amyloid fibril formation by α-synuclein. The fact that I_*max*_ and t_50_ did not always yield a consistent result strongly suggested that both EGCG and EGCG_*ox*_ could interfere with ThT fluorescence under some of the examined solution conditions. For example, EGCG seemed to inhibit the aggregation at pH 6 in a high-binding plate with glass beads, but the t_50_ and the soluble protein at the end of the measurement showed no effect on the amyloid formation. A possible mechanism of this observed effect on ThT fluorescence could either be direct interference [59] or competition with ThT for binding sites on the fibril surface [60]. It is therefore important to not rely on ThT intensity alone when assessing the inhibitory effects of EGCG, or indeed any other compound. However, our results showed that even under conditions where the absolute ThT fluorescence intensity was influenced by a potential inhibitor, careful experimental design and analysis of the shapes of the kinetic curves, rather than of the absolute intensity, allowed important mechanistic insight to be generated, in particular if the ThT experiments were combined with other experimental techniques. The inclusion of imaging techniques, such as AFM, could help to avoid false positives, but we also showed in this work that it could be challenging to obtain representative images of the content of an aggregated protein solution. We therefore employed an additional method in this work, MDS, that allowed us to quantify both the concentration and average size of the protein remaining in the supernatant of the completed aggregation reaction after centrifugation. This study revealed the strong influence of the de novo aggregation conditions on the mode of action of a given compound. In a non-binding surface plate, the presence of EGCG was observed to result in a faster kinetics of aggregation. Unlike the protein, the compound could bind to the protein-repellant surface and thereby enable the α-synuclein monomer to bind to the so-modified surface. This paved the way toward the nucleation of amyloid fibrils and started the amyloid cascade. Consistent with this model, pre-incubation of the wells of a non-binding plate could also lead to efficient induction of aggregation, and a clear concentration dependence of this effect was visible. A de novo experiment of α-synuclein amyloid fibril formation involved different microscopic steps. At neutral pH conditions, the process was dominated by primary nucleation, growth, and fragmentation. Secondary nucleation on the surface of preformed α-synuclein fibrils became more important, when the pH was decreased towards the isoelectric point. By performing seeded experiments, the investigation of the process of fibril elongation in isolation was feasible. In strongly seeded experiments (5% seeds in monomer equivalents), mostly EGCG_*ox*_ exerted an effect on the seeding efficiency, and this inhibitory effect appeared most pronounced at acidic pH values. At the lowest pH values, de novo aggregation, aided by the coating of the wells by EGCG, became efficient enough to lead to rapid aggregation despite the lack of seeding. More weakly seeded experiments (0.5% seeds in monomer equivalents) were more susceptible to inhibition, probably because the relative concentration ratio of inhibitor to seeds was higher. These seeded experiments provided clear evidence for an interaction between α-synuclein fibrils and both EGCG and EGCG_*ox*_ under most pH conditions. This was consistent with a previous study that reported an affinity of EGCG for α-synuclein fibrils in the low μM range at neutral pH [61]. Indeed, in this previous work, it was noted that the affinity of EGCG to α-synuclein fibrils appeared to become tighter over the course of minutes to hours. This corresponded to the time scale of EGCG oxidation under these conditions and was consistent with our finding here that EGCG_*ox*_ had a stronger inhibitory effect at equivalent concentrations. We also performed an experiment to probe whether EGCG_*ox*_ interacted with fibrils at more acidic pH than what had been shown in the previous study. When 25 μM α-synuclein fibrils was incubated with a stoichiometric quantity of EGCG_*ox*_, approximately two thirds of the compound could be centrifuged down with the fibrils. This result confirmed that α-synuclein fibrils interacted with EGCG_*ox*_ also at acidic pH with a stoichiometry not very different from 1:1.

Since we detected an inhibitory effect of EGCG_*ox*_ in seeded experiments, particularly in the experiments at low seed concentrations, we tested whether incubating the seeds with EGCG or EGCG_*ox*_ before the experiment was able to influence their seeding efficiency. In experiments where the seeds were incubated for 1 h and then added at a final concentration of 5% in monomer equivalents at pH 5–7, the aggregation kinetics and the efficiency of the seeds did not reveal any differences between pre-incubated seeds and those that had not been in contact with EGCG or EGCG_*ox*_. We then tested whether the seeds were altered by a substantially longer incubation (>100 h) at 37 °C, followed by addition of fresh monomer. Furthermore, here, we found no influence on the seeding efficiency, but the ThT fluorescence intensity was strongly decreased by the compounds. In these pre-incubation experiments, the concentrations of EGCG/EGCG_*ox*_ were sub-stoichiometric during the actual elongation reaction. Taken together, our seeded experiments showed therefore that EGCG and/or its oxidation products could only act on the seeding efficiency if present at high enough concentrations and that the interactions between the seeds and the compound were not able to alter the seeding efficiency permanently.

This finding agreed with a previous observation for κ-casein fibrils, which interacted with high affinity with EGCG, but showed no indication of the modification of the structure or of the redirection of the aggregation pathway [62]. However, various studies have reported the EGCG-induced remodeling of diverse amyloid fibrils and the formation of soluble amorphous aggregates at neutral pH [31,35,58,60,63,64,65,66,67]. Our seeding data did not suggest a seeding efficiency change or dissociation of the pre-formed α-synuclein fibrils; EGCG interacted with the fibril surface and therefore changed the interaction with ThT, as well as being able to interfere with the seeding if present at sufficiently high concentrations. However, an equimolar concentration was not sufficient to change the structure of preformed fibrils in such a way as to render them less efficient as seeds.

All together, our results painted a much more complex picture of the inhibitory effects of EGCG on the amyloid fibril formation by α-synuclein than what the available literature suggested. We summarize our findings in Figure 11. First of all, EGCG itself seemed to be very ineffective as an inhibitor, whereas its oxidation products were much more efficient [45]. This finding explained why in general, EGCG was a rather inefficient inhibitor in de novo experiments under pH conditions where the compound was stable, i.e., mildly acidic pH. Furthermore, EGCG could interact with the non-binding surfaces of plates and transform them into efficient surfaces for the heterogeneous primary nucleation of α-synuclein amyloid fibrils. Finally, EGCG_*ox*_ was able to interfere effectively with seeded aggregation if present at a high enough ratio with respect to the seeds. However, upon dilution and subsequent unbinding, the fibrils recovered their seeding efficiency. Our seeded experiments were carried out under quiescent conditions, where fibril fragmentation was negligible and no new ends were therefore generated, explaining the efficient inhibition. Under conditions of vigorous mechanical shaking, such as the ones we employed in our de novo experiments, the constant generation of new ends through primary nucleation and fragmentation [10] rendered the inhibitory effect of EGCG_*ox*_ much weaker. Our study represents the most detailed investigation of the inhibitory effects of EGCG on α-synuclein amyloid fibril formation. From our study emerged a complex picture of the effects of EGCG on α-synuclein amyloid fibril formation. These data suggested that an extensive and multi-dimensional characterization of potential amyloid fibril inhibitors is required in order to be able to conclude whether a given molecule is a promising inhibitor candidate.

## 5. Conclusions

In conclusion, we showed that EGCG only inhibited the amyloid fibril formation by α-synuclein under very specific conditions and that this compound could even act as an enhancer of amyloid fibril formation through facilitating heterogeneous primary nucleation. The oxidation products of EGCG were significantly more efficient inhibitory agents than the unmodified EGCG, but at the same time, they are also more efficient in inducing primary nucleation. This led to a complex interplay of the inhibitory and enhancing effects of EGCG and EGCG_*ox*_, the net effects of which depended on the pH of the solution, the presence or absence of seeds, as well as the type of reaction vessel and the general conditions of the aggregation reaction. Importantly, we also established that EGCG was not able to remodel α-synuclein into seed-incompetent structures. Taken together, our results highlighted the complexity of even such a supposedly well-established amyloid inhibitor as EGCG and established a detailed experimental strategy to evaluate the potential of a compound to interfere with amyloid fibril formation.

## Figures and Tables

**Figure 1 ijms-21-01995-f001:**
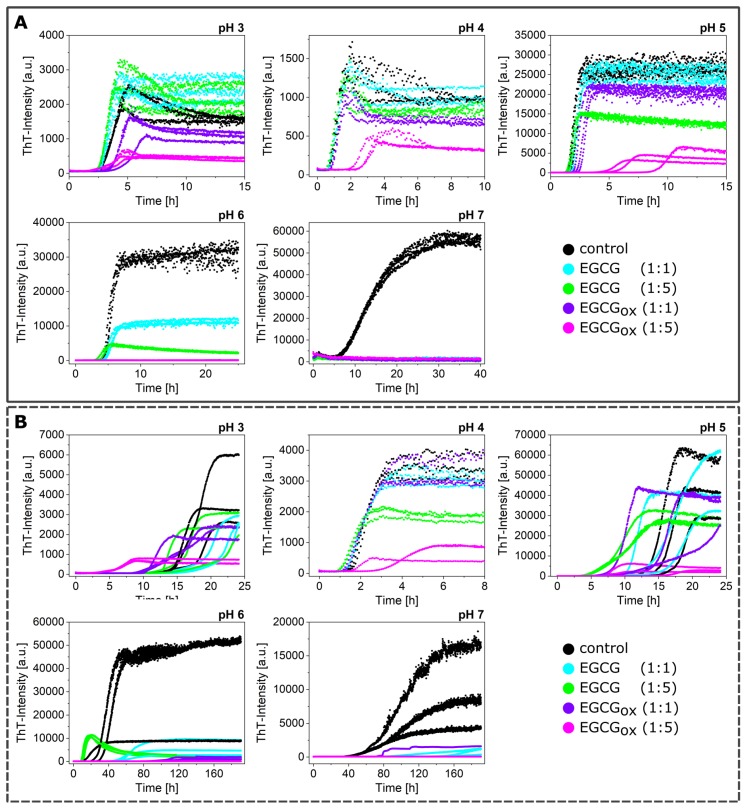
The effects of different ratios (1:1 and 5:1 with respect to protein) of EGCG and EGCG_*ox*_ on the aggregation kinetics of α-synuclein at different pH values (pH 3 to pH 7), monitored in high-binding surface plates in the presence (**A**) and absence (**B**) of glass beads.

**Figure 2 ijms-21-01995-f002:**
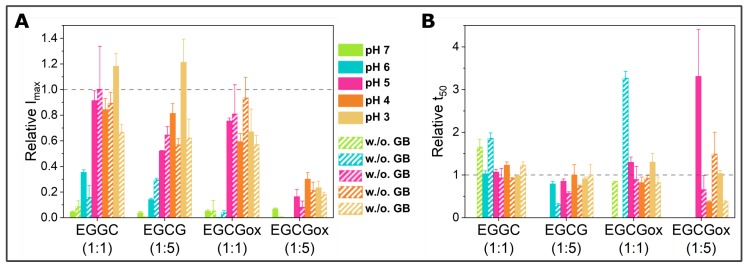
Overview of the effects of EGCG and EGCG_*ox*_ on α-synuclein aggregation monitored in high-binding surface plates assayed by (**A**) maximum ThT fluorescence intensity and (**B**) t_50_ of the aggregation time course. Filled bars represent aggregation in the presence of glass beads and striped bars in the absence of glass beads. Error bars are standard deviations. The data are normalized to the control of the corresponding condition, i.e., the aggregation in the absence of EGCG or EGCG_*ox*_, the kinetic parameters of which are indicated with the horizontal dashed line at a factor of one. Relative t_50_ values are only displayed, if any fibril formation is detected by an increase in ThT fluorescence intensity.

**Figure 3 ijms-21-01995-f003:**
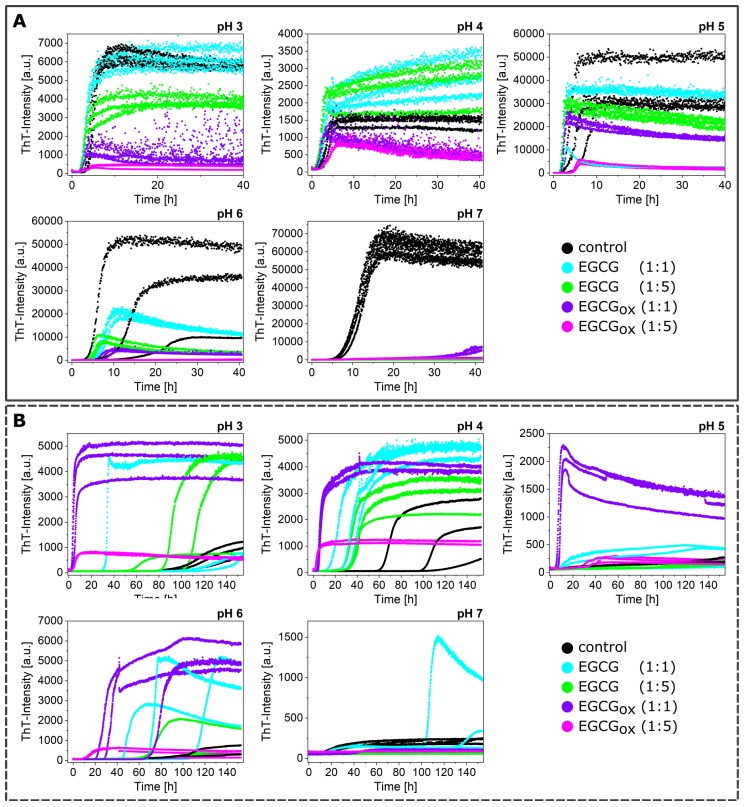
The effects of EGCG and EGCG_*ox*_ on the aggregation kinetics of α-synuclein at different pH values (pH 3 to pH 7) monitored in a non-binding surface plate in the presence (**A**) and absence (**B**) of glass beads.

**Figure 4 ijms-21-01995-f004:**
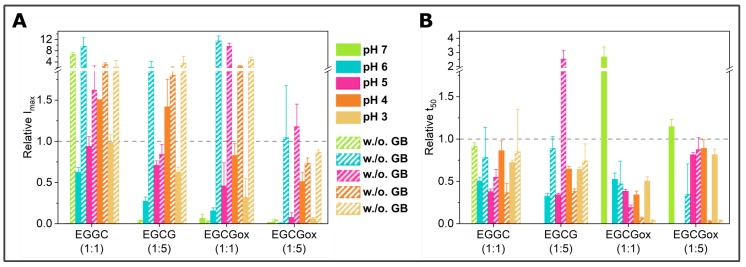
Overview of the effects of EGCG and EGCG_*ox*_ on α-synuclein aggregation monitored in a non-binding surface plate assayed by (**A**) maximum ThT fluorescence intensity and (**B**) t_50_ of the aggregation. Filled bars represent aggregation in the presence of glass beads and striped bars without glass beads. Error bars are standard deviations. The data are normalized to the control of the corresponding condition, and the comparison is outlined with the dashed line (at 1).

**Figure 5 ijms-21-01995-f005:**
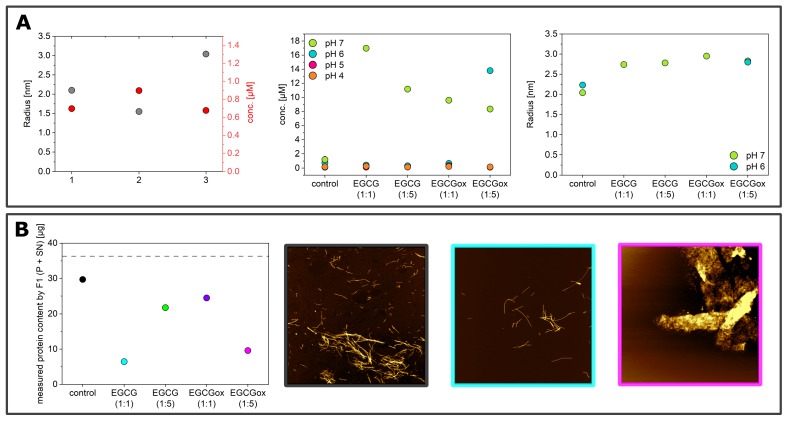
(**A**) Soluble α-synuclein concentration measured in the supernatant after centrifuging the end product of the aggregation reactions in a non-binding surface plate in the presence of glass beads. Radius in nm and concentration in μM of the three replicates of α-synuclein at pH 6 (control) (left), concentration in μM (middle), and radius in nm (right) of the end product of the aggregation reactions at pH 4, pH 5, pH 6, and pH 7. The three replicates per condition were combined before centrifugation (except for the control at pH 6, where each replicate sample was analyzed separately; see (**A**). (**B**) Amount of protein measurable with the Fluidity One (F1) MDS instrument (supernatant + pellet) in μg in the end-product of α-synuclein at pH 7 in a non-binding surface plate without additional glass beads (left). The dotted line indicates the used amount of protein. AFM height images of the control (black frame), α-synuclein with EGCG (1:1) (cyan frame) and of α-synuclein with EGCG_*ox*_ (1:5) (magenta frame). The image scale is 5 × 5 μM. The color range represents the height from −2 to 10 nm (left and middle) and −10 to 25 nm (right).

**Figure 6 ijms-21-01995-f006:**
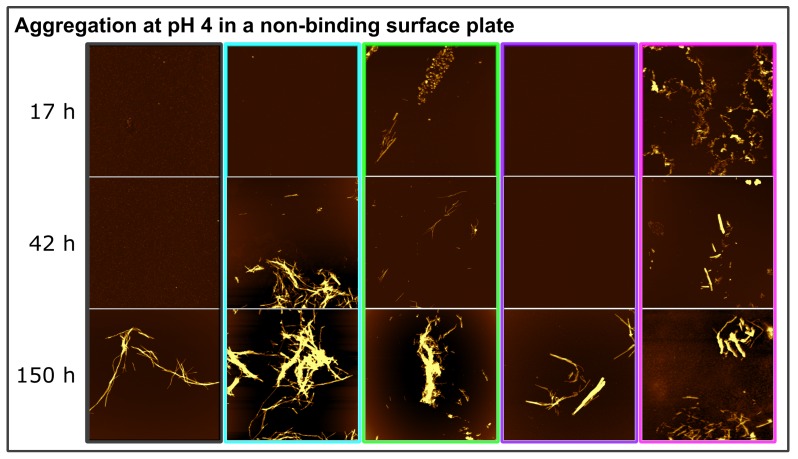
Time-resolved AFM height images of α-synuclein aggregation at pH 4 in a non-binding surface plate without glass beads. The colors of the frame correspond to the conditions (Figure 3): control (black frame), EGCG (1:1) (cyan frame), EGCG (1:5) (green frame), EGCG_*ox*_ (1:1) (purple frame), and EGCG_*ox*_ (1:5) (magenta frame). The image scale is 5 × 5 μM. The color range of the image represents the height range from −5 to 20 nm.

**Figure 7 ijms-21-01995-f007:**
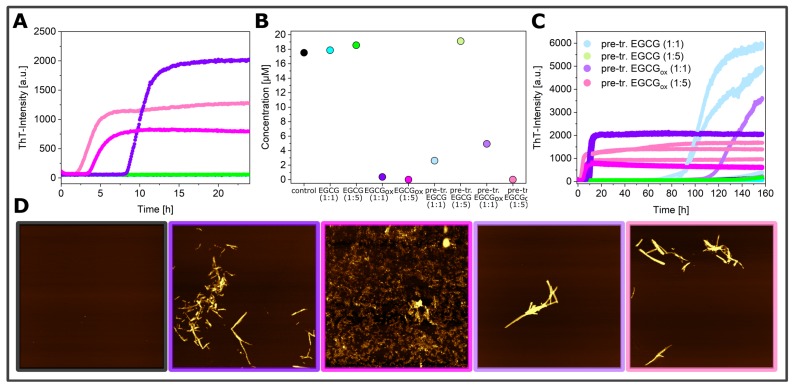
Aggregation kinetics of α-synuclein at pH 4 in a non-binding surface plate under quiescent conditions in the absence of glass beads. The fibril formation was monitored in the presence and absence of EGCG or EGCG_*ox*_ and in wells that were pre-treated with EGCG-solutions (**A**) and the corresponding concentration measurement by Fluidity One after 160 h (**B**) with AFM height images (**D**) of the aggregation products of α-synuclein (black frame) in the presence of EGCG_*ox*_ (1:1) (purple frame) and (1:5) (magenta frame), in the pre-treated wells with EGCG_*ox*_ (1:1) (light purple frame) and (1:5) (light magenta frame), and the overview of the three replicates per condition (**C**). The image scale is 5 × 5 μM. The color range represents the height from −3 to 12 nm.

**Figure 8 ijms-21-01995-f008:**
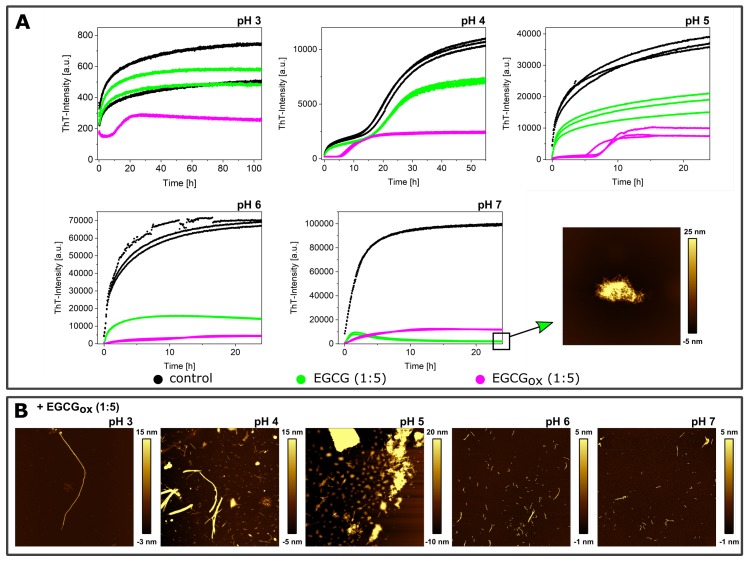
(**A**) The effects of EGCG and EGCG_*ox*_ on the aggregation kinetics of α-synuclein, in particular the growth of fibrils, at different pH values (pH 3 to pH 7) in the presence of 5% seeds monitored in a non-binding surface plate under quiescent conditions and a AFM height image of the sample at pH 7 in presence of EGCG (1:5) and (**B**) AFM height images of α-synuclein in the presence of EGCG_*ox*_ (1:5) at different pH values after the aggregation experiment. The image scale is 5 × 5 μM.

**Figure 9 ijms-21-01995-f009:**
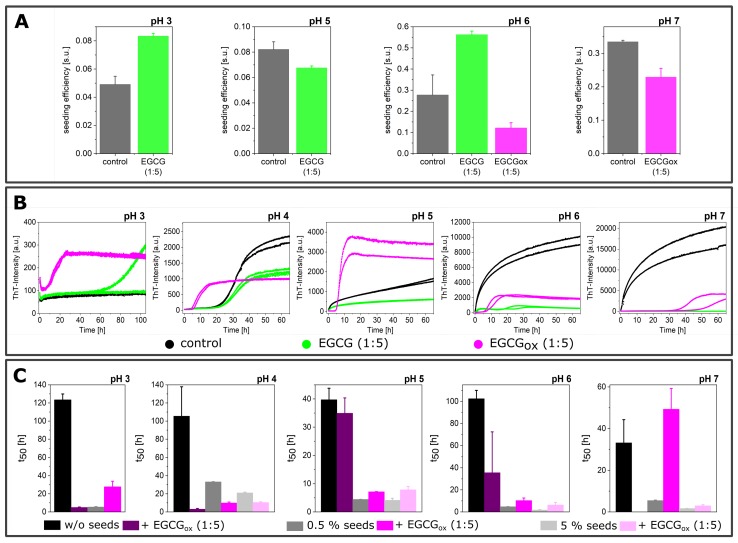
(**A**) The seeding efficiency, expressed in seeding units (s.u., [49]), determined by fitting the kinetics of the 5% seeding experiments with y = 1 − e^−*kt*^ after normalization between zero and one. Only the kinetics that showed the shape expected for a strongly seeded aggregation curve [49] were analyzed. (**B**) The effects of EGCG and EGCG_*ox*_ on the aggregation kinetics of α-synuclein at different pH values (pH 3 to pH 7) in the presence of 0.5% seeds monitored in a non-binding surface plate under quiescent conditions. (**C**) The t_50_ of α-synuclein at different pH values (pH 3 to pH 7) in a non-binding surface plate without additional seeds with shaking (black bar), with 0.5% seeds (dark grey bar) and 5% seeds (light grey bar) under quiescent conditions and in the presence of 1:5 EGCG_*ox*_ (violet).

**Figure 10 ijms-21-01995-f010:**
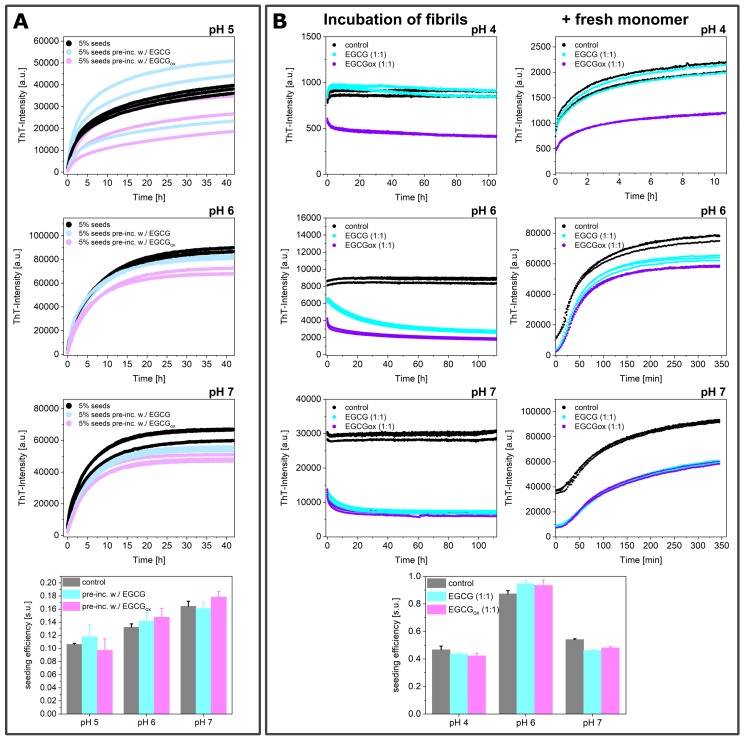
(**A**) The effects of EGCG and EGCG_*ox*_ on the seeded aggregation, when the seeds were pre-incubated with stoichiometric amounts of the compound for 2 h at RT before adding them to a 25 μM monomer-solution at pH 5, pH 6, and pH 7 to a final concentration of 5% (in monomer equivalents). The samples, where the fibrils were pre-incubated with the compound, contained still 1.25 μM EGCG or EGCG_*ox*_. (**B**) 10 μM fibrils at pH 4, pH 6, and pH 7 were incubated in the presence of 10 μM EGCG or EGCG_*ox*_ in a non-binding surface plate at 37 °C for over 100 h (left), and then, 50 μM fresh monomer was added (right).

**Figure 11 ijms-21-01995-f011:**
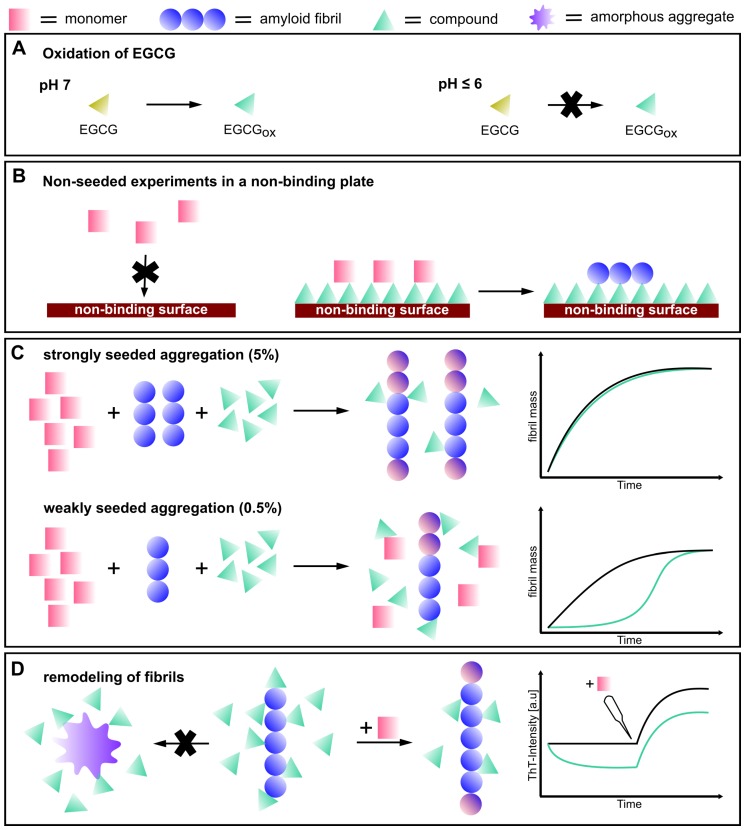
Schematic depiction of the effects of the compound EGCG on α-synuclein amyloid fibril formation. (**A**) EGCG oxidizes at pH 7, whereas it is stable at pH 6 and below. (**B**) illustrates that α-synuclein cannot bind to the non-binding surface of the multi-well plate, while in particular, EGCG_*ox*_ can bind to the surface and facilitate the formation of amyloid fibrils. (**C**) EGCG displayed almost no effect on a strongly seeded aggregation reaction (5% seeds), whereas a weakly seeded aggregation reaction (0.5% seeds) is inhibited more strongly. (**D**) The compound seems to interact with amyloid fibrils, but was not found to be able to remodel the fibrils into amorphous, seeding-incompetent aggregates. When fresh monomer was added, the fibrils had the same seeding efficiency as the control fibrils.

**Table 1 ijms-21-01995-t001:** Evaluation of the effects of EGCG and EGCG_*ox*_ on the de novo α-synuclein aggregation process established by comparing experimental values of I_*max*_ or t_50_ of the control samples (α-synuclein) with the ones determined in the presence of the component using one-way ANOVA (* *p* < 0.05; ** *p* < 0.01; *** *p* < 0.001). If the effect was defined as inhibitory without the indication of the *p*-value, the sample showed no aggregation during the term. The abbreviations HBS stands for high-binding surface, NBS for non-binding surface, and GB for glass bead.

**Assessed by Change in I_*max*_**						
		Conditions	EGCG (1:1)	EGCG (1:5)	EGCG_*ox*_ (1:1)	EGCG_*ox*_ (1:5)
physiologically relevant conditions	**pH 7**	HBS + GB	Inhibitory ***	Inhibitory ***	Inhibitory ***	Inhibitory ***
		HBS − GB	Inhibitory *	Inhibitory *	Inhibitory *	Inhibitory *
		NBS + GB	Inhibitory ***	Inhibitory ***	Inhibitory ***	Inhibitory ***
		NBS − GB	No effect	Inhibitory ***	Inhibitory ***	Inhibitory ***
	**pH 6**	HBS + GB	Inhibitory ***	Inhibitory ***	Inhibitory ***	Inhibitory ***
		HBS − GB	Inhibitory *	No effect	Inhibitory *	Inhibitory *
		NBS + GB	No effect	No effect	Inhibitory *	Inhibitory *
		NBS − GB	Enhancing **	No effect	Enhancing ***	No effect
	**pH 5**	HBS + GB	No effect	Inhibitory ***	Inhibitory ***	Inhibitory ***
		HBS − GB	No effect	No effect	No effect	Inhibitory **
		NBS + GB	No effect	No effect	No effect	Inhibitory ***
		NBS − GB	No effect	No effect	Enhancing ***	No effect
	**pH 4**	HBS + GB	No effect	No effect	Inhibitory *	Inhibitory ***
		HBS − GB	No effect	Inhibitory ***	No effect	Inhibitory ***
		NBS + GB	No effect	No effect	No effect	No effect
		NBS − GB	Enhancing **	No effect	Enhancing *	No effect
	**pH 3**	HBS + GB	No effect	No effect	No effect	Inhibitory ***
		HBS − GB	No effect	No effect	No effect	Inhibitory **
		NBS + GB	No effect	Inhibitory *	Inhibitory ***	Inhibitory ***
		NBS − GB	No effect	No effect	No effect	No effect
**Assessed by Change in t_50_**						
		Conditions	EGCG (1:1)	EGCG (1:5)	EGCG_*ox*_ (1:1)	EGCG_*ox*_ (1:5)
physiologically relevant conditions	**pH 7**	HBS + GB	Inhibitory	Inhibitory	Inhibitory	Inhibitory
		HBS − GB	Inhibitory **	Inhibitory	Inhibitory	Inhibitory
		NBS + GB	Inhibitory	Inhibitory	Inhibitory **	No effect
		NBS − GB	No effect	Inhibitory	No effect	Inhibitory
	**pH 6**	HBS + GB	No effect	No effect	Inhibitory *	Inhibitory
		HBS − GB	Inhibitory **	Enhancing *	Inhibitory ***	Inhibitory
		NBS + GB	No effect	No effect	No effect	Inhibitory
		NBS − GB	No effect	No effect	No effect	No effect
	**pH 5**	HBS + GB	No effect	No effect	No effect	Inhibitory **
		HBS − GB	No effect	No effect	No effect	No effect
		NBS + GB	Enhancing **	Enhancing **	Enhancing **	No effect
		NBS − GB	No effect	Inhibitory ***	Enhancing *	No effect
	**pH 4**	HBS + GB	No effect	No effect	No effect	Inhibitory***
		HBS − GB	No effect	No effect	No effect	No effect
		NBS + GB	No effect	Enhancing **	Enhancing ***	No effect
		NBS − GB	Enhancing **	Enhancing **	Enhancing ***	Enhancing ***
	**pH 3**	HBS + GB	No effect	No effect	Inhibitory *	No effect
		HBS − GB	No effect	No effect	No effect	Enhancing **
		NBS + GB	Enhancing **	Enhancing ***	Enhancing ***	Enhancing *
		NBS − GB	No effect	No effect	Enhancing **	Enhancing **

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
