# Peer review of "The Aggregation Conditions Define Whether EGCG is an Inhibitor or Enhancer of α-Synuclein Amyloid Fibril Formation"

_ijms, 2020, doi:10.3390/ijms21061995_

Round 1

Reviewer 1 Report

Sternke-Hoffman et al., examine in vitro the ability of Epigallocatechin gallate (EGCG) to affect amyloid fibril formation of a-synuclein. The authors use a variety of conditions (different pH conditions; non-binding versus binding surfaces; time; addition of glass beads) to assess these effects. Since there are so many conditions, it is difficult to follow the results at times, but the addition of Table 1 and Figure 11 help with the interpretation immensely. The experiments are thorough, and the manuscript is well-written.  I am concerned about this statement though “A small subset of this data (pH 6 andpH 7, 1:1 EGCG and 1:1 EGCGox) is from (Sneideris et al., 2019).” Data in the figure should not be published elsewhere without permission from the journal. I would suggest repeating the experiments to avoid this problem. In addition, the last paragraph of the introduction is particularly long.  I would suggest breaking it up into at least 2 paragraphs.

Author Response

Dear editor,

thank you very much for considering our manuscript for publication in the special issue on natural product inhibitors of amyloid fibril formation. Please find below a point by point reply to the issues raised by the reviewer 1.

Reviewer 1:

Comments and Suggestions for Authors

Sternke-Hoffman et al., examine in vitro the ability of Epigallocatechin gallate (EGCG) to affect amyloid fibril formation of a-synuclein. The authors use a variety of conditions (different pH conditions; non-binding versus binding surfaces; time; addition of glass beads) to assess these effects. Since there are so many conditions, it is difficult to follow the results at times, but the addition of Table 1 and Figure 11 help with the interpretation immensely. The experiments are thorough, and the manuscript is well-written.

Reply: We thank the reviewer for his/her positive overall evaluation of our work.

I am concerned about this statement though “A small subset of this data (pH 6 and pH 7, 1:1 EGCG and 1:1 EGCGox) is from (Sneideris et al., 2019).” Data in the figure should not be published elsewhere without permission from the journal. I would suggest repeating the experiments to avoid this problem.

Reply: We appreciate this view of the reviewer. We have repeated the relevant experiments, and we have replaced the old data by the new ones.

In addition, the last paragraph of the introduction is particularly long.  I would suggest breaking it up into at least 2 paragraphs.

Reply: We have divided this paragraph into several shorter ones.

Reviewer 2 Report

The manuscript ” The aggregation conditions define whether 2 EGCG is an inhibitor or enhancer of α-synuclein amyloid fibril formation” by Rebecca Sternke-Hoffmann, Alessia Peduzzo, Najoua Bolakhrif, Rainer Haas and Alexander K. Buell addresses very timely an important issue concerning the biological activity of polyphenolic compound EGCG in preventing the amyloid aggregation of α-synuclein – protein involved in the amyloid formation and depositions in Parkinson’s and other synucleinopathies. This issue is particularly important as EGCG is commonly viewed as an inhibitor of amyloid formation of various polypeptides, yet the careful and detail analysis of its mechanisms of action was missing. The authors conducted very extended and consistent examination of various conditions and convincingly demonstrated that EGCG inhibits the amyloid formation of α-synuclein only under very specific conditions, but can act also as an enhancer of amyloid fibril formation through facilitating heterogeneous primary nucleation. The oxidation product of EGCG was shown to be significantly more efficient inhibitory agent than the unmodified EGCG, however it proved to be also more efficient in inducing primary nucleation. Thus a complex interplay of the inhibitory and enhancing effects of EGCG and EGCGox was demonstrated, the outcome of which depends on the pH of the solution, the presence or absence of seeds, the type of vessel in which reaction is conducted and the general conditions of the aggregation reaction. The authors have also establish that EGCG is not able to remodel α-synuclein into seed-incompetent structures, which is very important in one would aim at complete blocking amyloid self-assembly. All data together is very consistent and indicates that even established amyloid inhibitor as EGCG may display more complex effects on the amyloid formation process and should be used with care. In general the authors have outlined the scheme how the compound in question should be tested in order to draw the conclusions on its amyloid preventing potency. The research is clearly presented, including both text and figures. The conclusions are well discussed. Therefore, taken all together the manuscript is recommended for publication without reservations.

Author Response

Dear editor,

thank you very much for considering our manuscript for publication in the special issue on natural product inhibitors of amyloid fibril formation. Please find below a point by point reply to the issues raised by the reviewer 2.

Reviewer: The manuscript ” The aggregation conditions define whether EGCG is an inhibitor or enhancer of α-synuclein amyloid fibril formation” by Rebecca Sternke-Hoffmann, Alessia Peduzzo, Najoua Bolakhrif, Rainer Haas and Alexander K. Buell addresses very timely an important issue concerning the biological activity of polyphenolic compound EGCG in preventing the amyloid aggregation of α-synuclein – protein involved in the amyloid formation and depositions in Parkinson’s and other synucleinopathies. This issue is particularly important as EGCG is commonly viewed as an inhibitor of amyloid formation of various polypeptides, yet the careful and detail analysis of its mechanisms of action was missing. The authors conducted very extended and consistent examination of various conditions and convincingly demonstrated that EGCG inhibits the amyloid formation of α-synuclein only under very specific conditions, but can act also as an enhancer of amyloid fibril formation through facilitating heterogeneous primary nucleation. The oxidation product of EGCG was shown to be significantly more efficient inhibitory agent than the unmodified EGCG, however it proved to be also more efficient in inducing primary nucleation. Thus a complex interplay of the inhibitory and enhancing effects of EGCG and EGCGox was demonstrated, the outcome of which depends on the pH of the solution, the presence or absence of seeds, the type of vessel in which reaction is conducted and the general conditions of the aggregation reaction. The authors have also establish that EGCG is not able to remodel α-synuclein into seed-incompetent structures, which is very important in one would aim at complete blocking amyloid self-assembly. All data together is very consistent and indicates that even established amyloid inhibitor as EGCG may display more complex effects on the amyloid formation process and should be used with care. In general the authors have outlined the scheme how the compound in question should be tested in order to draw the conclusions on its amyloid preventing potency. The research is clearly presented, including both text and figures. The conclusions are well discussed. Therefore, taken all together the manuscript is recommended for publication without reservations.

Reply: We thank the reviewer for his/her very positive evaluation of our work.

Reviewer 3 Report

This study reports important observations. Namely the possibility of inhibiting the growth of alpha-synuclein amyloid fibrils using the polyphenol epigallocatechin-3-gallate (EGCG) found in green tea. The main idea of the work is that EGCG can both inhibit and enhance amyloid fibril growth depending on aggregation conditions. However, I think that before publication some points need to be addressed:

  1. Practically all experiments were done both with EGCG and its auto-oxidised form (EGCGox). In this connection, a number of questions arise:
    a) In the Introduction it is not clearly explained why it is important. It is only said that “..at physiological pH, EGCG is unstable and auto-oxidises..” (line 80). It is not clear why experiments with EGCGox were done for all pH.
    b) It is also not clear why experiments with EGCG were done at pH 7 and pH 6 if in these conditions it is very unstable and oxidises rapidly “in a similar time scale as the aggregation process itself” (lines 460-463).
    c) EGCGox was used in all experiments and it also has a significant effect on the growth of amyloid fibrils, but it is not mentioned either in the title, in the Abstract, or in the final scheme (Fig. 11). Why?
    d) The explanation of the abbreviation EGCGox must be given when it firstly appeared.

  1. The authors probed the formation of amyloid fibrils formation with ThT fluorescence. This, of course, is a good method, but at the end of article it turned out that it is not suitable in this work:
    Line 468: “both EGCG and EGCGox can interfere with ThT fluorescence“
    Lines 485488: “In particular the oxidised EGCG strongly quenches ThT fluorescence, rather than inhibiting amyloid fibril growth. The compound either interferes with ThT-fluorescence (Buell et al., 2010) or binds to amyloid fibrils, preventing the binding of ThT (Palhano et al., 2013; Sneideris et al., 2019).”
    It means that the performed experiments do not carry any information. It is very good that the authors themselves drew attention to this. It is clear that a huge experimental work has been done and it's hard for authors to part with it. However, in the article it is necessary to give only experiments that carry important information on the basis of which reliable conclusions can be done. The authors could write in the Introduction that fluorescence of ThT is a wonderful method, but in this case it is not suitable and explain why. And that’s all. All unnecessary experimental data that only confuse the reader must be removed.

  1. In the Introduction the authors wrote “Once formed, amyloid aggregates are often highly stable ...” (line 54). This is a really widespread and well-established point of view. However, a publication has recently appeared in which the opposite opinion is expressed (Sulatsky  et al, 2020. PMID: 32057863). It would be good to discuss it.

Author Response

Dear editor,

thank you very much for considering our manuscript for publication in the special issue on natural product inhibitors of amyloid fibril formation. Please find below a point by point reply to the issues raised by the reviewer 3.

Reviewer 3:

Comments and Suggestions for Authors

This study reports important observations. Namely the possibility of inhibiting the growth of alpha-synuclein amyloid fibrils using the polyphenol epigallocatechin-3-gallate (EGCG) found in green tea. The main idea of the work is that EGCG can both inhibit and enhance amyloid fibril growth depending on aggregation conditions. However, I think that before publication some points need to be addressed:

1) Practically all experiments were done both with EGCG and its auto-oxidised form (EGCGox). In this connection, a number of questions arise:

a) In the Introduction it is not clearly explained why it is important. It is only said that “..at physiological pH, EGCG is unstable and auto-oxidises..” (line 80). It is not clear why experiments with EGCGox were done for all pH.

Reply: We thank the reviewer for pointing out that we have not made it sufficiently clear why experiments under all conditions were performed with both EGCG and EGCGox. Knowing that EGCG is unstable at neutral pH, our aim was to find out whether EGCG itself or its oxidation product is the more potent inhibitor. This can only be tested under conditions where EGCG itself is stable during the time course of the aggregation reaction, i.e. pH 6 and below. It was therefore also necessary to test the effect of EGCox under these conditions, which we did by adding pre-oxidized EGCG. Even though EGCG does not get oxidized under mildly acidic conditions, it may experience different pH environments inside an organism treated with the compound, ranging from neutral to mildly acidic (i.e. in lysozomes), similarly to the asyn itself. We therefore deem it relevant to study also the effects of EGCGox under acidic conditions. We have attempted to make this reasoning clearer in the manuscript.

b) It is also not clear why experiments with EGCG were done at pH 7 and pH 6 if in these conditions it is very unstable and oxidises rapidly “in a similar time scale as the aggregation process itself” (lines 460-463).

Reply: These solution conditions correspond to the most widely studied conditions, in particular the inhibitory effect of EGCG on asyn amyloid fibril formation has first been reported under these conditions. However, previous studies have not generally considered the fact that simultaneously with the aggregation reaction, the EGCG undergoes chemical modifications. We therefore think it is important to also revisit these well-studied conditions with our fresh perspective.

c) EGCGox was used in all experiments and it also has a significant effect on the growth of amyloid fibrils, but it is not mentioned either in the title, in the Abstract, or in the final scheme (Fig. 11). Why?

Reply: We agree with the reviewer that EGCGox should feature more prominently, given that it has both a stronger inhibitory and enhancing effect on asyn aggregation compared to fresh EGCG. We had initially thought to keep things simpler by not always mentioning both EGCG and EGCGox, but we now mention EGCGox in the abstract as well as in the final summary figure.

d) The explanation of the abbreviation EGCGox must be given when it firstly appeared.

Reply: We have made this change.

2) The authors probed the formation of amyloid fibrils formation with ThT fluorescence. This, of course, is a good method, but at the end of article it turned out that it is not suitable in this work:
Line 468: “both EGCG and EGCGox can interfere with ThT fluorescence“
Lines 485488: “In particular the oxidised EGCG strongly quenches ThT fluorescence, rather than inhibiting amyloid fibril growth. The compound either interferes with ThT-fluorescence (Buell et al., 2010) or binds to amyloid fibrils, preventing the binding of ThT (Palhano et al., 2013; Sneideris et al., 2019).”
It means that the performed experiments do not carry any information. It is very good that the authors themselves drew attention to this. It is clear that a huge experimental work has been done and it's hard for authors to part with it. However, in the article it is necessary to give only experiments that carry important information on the basis of which reliable conclusions can be done. The authors could write in the Introduction that fluorescence of ThT is a wonderful method, but in this case it is not suitable and explain why. And that’s all. All unnecessary experimental data that only confuse the reader must be removed.

Reply: We have probably not expressed ourselves clearly enough in this respect and we thank the reviewer for pointing this out. However, we strongly disagree with the statement that the ThT experiments in general carry no information. This is clearly not the case, as we have been able to draw a large amount of mechanistic information from our ThT experiments. While EGCG can indeed quench ThT fluorescence, we find that under most conditions, this quenching effect is incomplete and that ThT is still able to indicate the formation of fibrils, albeit with lower fluorescence quantum yield. In our revised version we have stressed even more clearly that ThT fluorescence intensity alone can be rather unreliable as a readout for the evaluation of the inhibitory effects of a compound. If, however, the lag time, half time or overall shape of a kinetic curve is significantly altered, such as in many of our experiments, an (inhibitory or enhancing) effect can be concluded, whether ThT intensity is affected or not. Only in the cases where no ThT fluorescence is observed at all (de novo aggregation at pH 7), the use of other techniques such as imaging or microfluidic diffusional sizing, is indispensable, which we have done. We have attempted to express these thoughts more clearly in the manuscript.

3) In the Introduction the authors wrote “Once formed, amyloid aggregates are often highly stable ...” (line 54). This is a really widespread and well-established point of view. However, a publication has recently appeared in which the opposite opinion is expressed (Sulatsky et al, 2020. PMID: 32057863). It would be good to discuss it.

Reply: We thank the reviewer for pointing this very recent publication out to us. We have modified our statement on the stability of amyloid fibrils and added a citation to this publication.

Round 2

Reviewer 1 Report

The authors have addressed my concerns.

Reviewer 3 Report

The authors reply to all my comments and improved the manuscript and figure 11. The authors did not convince me that ThT fluorescence data are very informative. However, I believe that the article can be published as is.